Multiplexed ISSR genotyping by sequencing distinguishes two precious coral species (Anthozoa: Octocorallia: Coralliidae) that share a mitochondrial haplotype

Takata Kenji 1 2
Taninaka Hiroki 3
Nonaka Masanori 4
Iwase Fumihito 5
Kikuchi Taisei 6
Suyama Yoshihisa 7
Nagai Satoshi 8
Yasuda Nina nina27@cc.miyazaki-u.ac.jp 2
1 Graduate School of Agriculture, Faculty of Agriculture, University of Miyazaki , Miyazaki , Miyazaki , Japan
2 Department of Marine Biology and Environmental Sciences, Faculty of Agriculture, University of Miyazaki , Miyazaki , Miyazaki , Japan
3 Interdisciplinary Graduate School of Agriculture and Engineering, University of Miyazaki , Miyazaki , Miyazaki , Japan
4 Okinawa Churashima Foundation Reseach Center , Motobu , Okinawa , Japan
5 Shikoku Marine Life Laboratory , Otsuki , Kochi , Japan
6 Parasitology, Faculty of Medicine, University of Miyazaki , Miyazaki , Miyazaki , Japan
7 Field Science Center, Graduate School of Agricultural Science, Tohoku University , Osaki , Miyagi , Japan
8 National Research Institute of Fisheries Science, Japan Fisheries Research and Education Agency , Yokohama , Kanagawa , Japan
Rodriguez-Lanetty Mauricio
Electronic publication date: 2019 Oct 4
Publication date: 2019
Volume: 7
Electronic Location ID: e7769
Received 2019 Apr 24; Accepted 2019 Aug 27
Copyright: ©2019 Takata et al.
Copyright year: 2019
Copyright holder: Takata et al.
License: This is an open access article distributed under the terms of the Creative Commons Attribution License, which permits unrestricted use, distribution, reproduction and adaptation in any medium and for any purpose provided that it is properly attributed. For attribution, the original author(s), title, publication source (PeerJ) and either DOI or URL of the article must be cited.
License URL: https://creativecommons.org/licenses/by/4.0/

Keywords: MIG-seq, Incomplete lineage sorting, Species delimitation, Deep sea coral, Octocoral

Funding: Japan Society for the Promotion of Science (JSPS) 17H04996 16H04722 Grant-in-Aid for Research Fellows 201921342 Environmental Research Projects Program to Disseminate Tenure Tracking System from the Ministry of Education, Culture, Sports, Science and Technology (MEXT) Japan Society for the Promotion of Science (JSPS) KAKENHI 16H04722 This study was supported by the Japan Society for the Promotion of Science (JSPS) Grant-in-aid for young scientists (A) (17H04996),16H04722, a Grant-in-Aid for Research Fellows (201921342), and the Sumitomo Foundation Fiscal 2018 Grant for Environmental Research Projects and the Program to Disseminate Tenure Tracking System from the Ministry of Education, Culture, Sports, Science and Technology (MEXT). Additional support was granted by the Japan Society for the Promotion of Science (JSPS) KAKENHI (Grant Number 16H04722). The funders had no role in study design, data collection and analysis, decision to publish, or preparation of the manuscript.

==============================
Background

Precious corals known as coralliid corals (Anthozoa: Octocorallia) play an important role in increasing the biodiversity of the deep sea. Currently, these corals are highly threatened because of overfishing that has been brought on by an increased demand and elevated prices for them.The deep sea precious corals Pleurocorallium elatius and P. konojoi are distributed in Japanese waters and have distinct morphological features: (1) the terminal branches of the colony form of P. elatius are very fine, while those of P. konojoi are blunt and rounded, (2) the autozooids of P. elatius are arranged in approximately four rows, while those of P. konojoi are clustered in groups. However, previous genetic analysis using mtDNA and nuclear DNA did not indicate monophyly. Therefore, it is important to clarify their species status to allow for their conservation.

Methodology

We collected a total of 87 samples (60 of Corallium japonicum and 27 of P. konojoi) from around the Ryukyu Islands and Shikoku Island, which are geographically separated by approximately 1,300 km. We used a multiplexed inter-simple sequence repeat (ISSR) genotyping by sequencing (MIG-seq) and obtained 223 SNPs with which to perform STRUCTURE analysis and principle coordinate analysis (PCoA). In addition, two relatively polymorphic mtDNA regions were sequenced and compared.

Results

P. elatius and P. konojoi share a same mtDNA haplotype, which has been previously reported. However, MIG-seq analysis clearly distinguished the two species based on PCoA and STRUCTURE analysis, including 5% of species-specific fixed SNPs.

Conclusion

This study indicated that P. elatius and P. konojoi are different species and therefore both species should be conserved separately. Our findings highlight the importance of the conservation of these two species, especially P. elatius, whose population has been dramatically depleted over the last 100 years. The study also demonstrated the effectiveness and robustness of MIG-seq for defining closely related octocoral species that were otherwise indistinguishable using traditional genetic markers (mtDNA and EF).

Introduction

Precious corals belong to family Coralliidae (Anthozoa: Octocorallia) play an important role in increasing the biodiversity of the deep sea by providing a complex habitat structure, suitable for other species to live in Roberts, Wheeler & Freiwald (2006). Their beautiful red and pink axis has drawn great attention and they have been harvested for ornaments, jewelry, and currency since ancient times (Chen, 2012; Clark & Rowden, 2009). Recent studies, however, indicated precious corals are vulnerable to overfishing because of their low fecundity and slow growth rate (Nonaka, Nakamura & Muzik, 2015; Luan et al., 2013; Torrents et al., 2005). Consequently, there is an increasing need for the conservation of overexploited precious corals in order to avoid local extinction (CITES-Netherlands, 2007; CITES-Qatar, 2009).

Species delimitation is important for conserving precious corals due to the fact that the species is the most fundamental unit for preservation (De Queiroz, 2007). However, species identification of the precious corals is sometimes difficult, especially for closely related species. The use of morphological characteristics in determining the species identification of octocorals, including precious corals, is the typical method; however, variations, and high frequency of homoplasy sometimes hinder identification (Figueroa & Baco, 2014). More recently, molecular and phylogenetic approaches using partial mitochondrial DNA (mtDNA) have emerged as useful tools to identify octocoral species (McFadden et al., 2011; Pante et al., 2012; Quattrini et al., 2014). In Coralliidae, the intergenic region 1 (IGR1) in mtDNA has a relatively high intra-species variation and is proposed as a useful marker for distinguishing between precious coral species (Tu, Dai & Jeng, 2015). Although mtDNA is widely used for species delimitation, discordances with the morphological characteristics of species still remain. For example, morphologically distinct species in Hemicorallium cannot be delineated using mtDNA (Ardila, Giribet & Sánchez, 2012; Tu, Dai & Jeng, 2015). Indeed, the slow mtDNA evolution has long been recognized for anthozoans (Hellberg, 2006; France & Hoover, 2002). Octocoral species in particular have a slow mutation rate of mtDNA, due to octocoral-specific DNA repairing enzymes (Bilewitch & Degnan, 2011; France & Hoover, 2002). These facts indicate that a higher resolution genetic marker is required to elucidate the species status of closely related precious coral species.

The advent of high-throughput sequencing technology has rendered a large number of loci of non-model organisms accessible in a short time period. For example, the restriction site associated DNA sequence (RADseq) (Baird et al., 2008) has been applicable in non-model organisms to delimit closely related species of many reference taxa. In deep sea octocorals, only a few studies have applied RADseq for species delimitation. RADseq has helped fine-tune the species status of genus Chrysogorgia (Pante et al., 2014) and revealed robust species boundaries in the genus Paragorgia using the Bayesian model based-method (Herrera & Shank, 2016). Additional studies using the high-throughput technology to delineate closely related precious coral species would be helpful to reveal species boundaries.

Multiplexed inter simple sequence repeat (ISSR) genotyping by sequencing (MIG-seq) is an easy, cost-effective, novel method to obtain a moderate number of single nucleotide polymorphisms (SNPs) of non-model organisms using polymerase chain reaction (PCR) and high-throughput technology. The number of available SNPs from MIG-seq analysis is generally less than those using other techniques such as RAD-seq (Miller et al., 2007). However, MIG-seq has several advantages, namely that putatively neutral loci adjacent to microsatellite regions can be obtained and the method can be performed with small amounts and/or low-quality DNA, and is relatively easy to perform cheaply. The ISSR regions are first amplified by PCR using universal primer sets, indexed by a second PCR, and then sequenced to obtain up to a few thousand SNPs. Closely related species of Heliopora were successfully delineated using MIG-seq (Richards et al., 2018), implying the effectiveness of this method for other closely related octocoral species.

Our target species, deep sea precious corals P. elatius and P. konojoi are distributed in Japanese waters at depths of 115–330 m (Nonaka & Muzik, 2009). P. elatius and P. konojoi can be distinguished by at least two morphological features: (1) terminal branches of the colony form of P. elatius are fine, while those of P. konojoi are blunt and rounded, (2) autozooids of P. elatius are arranged in approximately four rows, while those of P. konojoi are clustered in groups (Nonaka et al., 2012). Nevertheless, a previous molecular analysis suggested the two species shared a major haplotype at eight mtDNA regions and EF-1, indicating that the species status of these two is unclear (Tu, Dai & Jeng, 2015). Because these two species, especially P. elatius, are facing the risk of local extinction due to overfishing (Nonaka & Muzik, 2009), clarifying whether or not they are the same species with gene flow occurring between them is important for devising conservation strategies.

In this study, we applied MIG-seq analysis to clarify the species status of the two closely related precious corals, P. elatius and P. konojoi in order to obtain basic information for the conservation of precious coral species. We also sequenced widely used mtDNA regions for comparison with the MIG-seq results.

Materials and Methods

Sample collection and DNA extraction

In total, 87 samples of P. elatius and P. konojoi (27 P. elatius and 60 P. konojoi) were collected from depths of 100 to 330 m using either a traditional coral net, underwater remotely operated vehicles (ROV), or submarines off Ryukyu, Kyushu, and Shikoku, Japan (Table 1). We collected some samples under the Kochi Prefecture sampling permit number Sa 401, Sa 412, and Sa 423 in Kochi. Samples were preserved in 90% ethanol and genomic DNA was extracted using the hot alkaline solution method (Meeker et al., 2007) followed by ethanol precipitation (Richards et al., 2018).

Table 1 Precious coral samples used in this study.

	Location		Coordinate	MIG-seq	mtDNA	Sampling year	Sampling method	
Pleurocorallium konojoi	Shikoku Island	Kouchi	32°30′41.87″N 132°49′42.26″E	47	0	Apr.11-Nov.14	traditional coral net	
	Ryukyu Islands	Iou Is.	30°47′31.812″N 132°18′13.5″E	5	4	Dec.07-Aug.09	manned submarine	
		Take Is.	30°48′36.684″N 130°25′43.356″E	7	7	Feb.08-May.09	manned submarine	
		Tanega Is.	30°36′34.38″N 130°58′43.932″E	1	1	Dec.07-Aug.08	manned submarine	
				60	12			
Pleurocorallium elatius	Shikoku Island	Kouchi	32°30′41.87″N 132°49′42.26″E	1	0	Jul.11	traditional coral net	
	Ryukyu Islands	Iou Is.	30°47′31.812″N 132°18′13.5″E	2	4	Jun.05-Apr.09	manned submarine	
		Take Is.	30°48′36.684″N 130°25′43.356″E	2	0	Sep.07-Oct.08	manned submarine	
		Tanega Is.	30°36′34.38″N 130°58′43.932″E	3	3	Feb.06-Dec.07	manned submarine	
		Yaku Is.	30°20′40.524″N 130°30′45.756″E	4	1	Aug.07-Aug.09	manned submarine	
		Amami Is.	28°16′4.224″N 129°21′43.596″E	1	0	Aug.07	manned submarine	
		Tokuno Is.	27°47′25.512″N 128°58′0.48″E	2	0	Oct.07-Aug.09	manned submarine	
		Izena Is.	26°56′2.976″N 127°56′28.932″E	1	1	Jan.06	ROV	
		Hateruma Is.	24°11′40.272″N 123°33′45.54″E	3	0	Sep.07-Mar.08	ROV	
		Ikema Is.	24°55′49.836″N 125°14′42.216″E	1	1	May.09	ROV	
		Tarama Is.	24°39′19.908″N 124°41′48.12″E	1	1	Aug.05-Dec.07	ROV	
		Ishigaki Is.	24°24′23.04″N 124°10′31.584″E	5	1	Nov.05-Mar.09	ROV	
		Nakanokami Is.	24°11′40.272″N 123°33′45.54″E	1	1	Mar.08-Apr.08	ROV	
				27	13			
Total				87	25			

MtDNA analysis

Two mtDNA regions were sequenced for P. elatius and P. konojoi: IGR1 region using primers (IGR1-Co-F and IGR1-Co-R, Tu, Dai & Jeng, 2015) and putative mitochondrial DNA mismatch repairing gene (mitochondrial mutS-like protein) regions (MSH-Co-F and MSH-Co-R, Tu, Dai & Jeng, 2015) following the original protocols. Two regions were directly sequenced from both directions using Big Dye v 3.1 and the Abi3730 sequencer. All of the sequence data were manually checked using Bioedit ver. 7.0.9.0 (Hall, 1999) and then all the sequences were aligned on GENETIX ver. 12. The two mtDNA region sequences were concatenated and used for subsequent analysis. To assess genetic diversity, the number of haplotypes, number of polymorphic sites, haplotype diversity (h) (Nei, 1987), and nucleotide diversity (Π) were calculated using DNaSP ver. 5.1 (Librado & Rozas, 2009). A median-joining network (Bandelt, Forster & Rohl, 1999) was constructed using NETWORK ver. 5.003, including gap sites, to visualize the evolutionary relationships between haplotypes obtained from P. elatius and P. konojoi. A maximum likelihood tree using MEGA 7.0 (Kumar, Stecher & Tamura, 2016) was constructed using concatenated mtDNA sequences. The best model was estimated using MEGA 7.0 based on the corrected Akaike Information Criterion and the Tamura 3-parameter model (Tamura, 1992) was used to construct a phylogenetic tree. Confidence values for phylogenetic trees were inferred using 1,000 bootstrap replicates. All mtDNA sequences obtained in this study were deposited (DDBJ Accession number: LC464485-LC464516, LC475110-LC475134).

MIG-seq analysis

We performed a MIG-seq analysis to detect genome-wide SNPs following the protocol by Suyama & Matsuki (2015). Briefly, MIG-seq amplifies putatively neutral, anonymous genome-wide ISSR regions (Gupta et al., 1994; Zietkiewicz, Rafalski & Labuda, 1994), including a few hundred to a few thousand SNPs, using 8 pairs of multiplex ISSR primers (MIG-seq primer set 1) for the first PCR. Then the DNA libraries from each sample with a different index were pooled and sequenced using MiSeq (sequencing control software v2.0.12, Illumina) with the MiSeq Reagent v3 150 cycle kit (Illumina). Image analysis and base calling were performed using real-time analysis software v1.17.21 (Illumina). We analyzed a total of 87 individuals with 27 P. elatius and 60 P. konojoi individuals collected from the Ryukyu Islands to Shikoku Island, geographically separated by approximately 1,300 km (Table 1).

To eliminate low-quality reads and primer sequences from the raw data, we used the FASTX-toolkit version 0.0.14 (fastaq_quality_filter) (Gordon & Hannon, 2012; http://hannonlab.cshl.edu/fastx_toolkit/index.html) with a fastq-quality-filter setting of –Q 33 –q 30 –p 40. We removed adapter sequences for the Illmina MiSeq run from both the 5’ end (GTCAGATCGGAAGAGCACACGTCTGAACTCCAGTCAC) and the 3′end (CAGAGATCGGAAGAGCGTCGTGTAGGGAAAGAC) using Cutadapt version 1.13 (Martin, 2011), and then we excluded short reads less than 80 bp. The quality-filtered sequence data were demultiplexed and filtered through the software Stacks v1.46 (Catchen et al., 2011; Catchen et al., 2013). We used Stacks v. 1.4 (Catchen et al., 2013) to stack the reads and extract SNPs. First, we used the U-stacks with the option settings of ‘minimum depth of coverage required to create a stack (m)′ = 3, ‘maximum distance allowed between stacks (M)′ = 1, ‘maximum distance allowed to align secondary reads to primary stacks (N)′ = 1, and the deleveraging (d) and removal (r) algorithms enabled. Secondly, we used the C-stacks with the option ‘number of mismatches allowed between sample loci when build the catalog (n)′ = 4, followed by the S-stacks. To confirm the consistency of the results using different sets of SNPs with different amounts of missing data, we first created three different SNP sets using population software implemented in Stacks v. 1.4 by restricting the data analysis to different criteria: (i) the minimum percentage of individuals required to process a locus across all data (r) was set at 70% and restricting data analysis to a single SNP per locus, (ii) r was set at 50% and a single SNP per locus was used, and (iii) r was set at 50% and all SNPs per locus were used. Confirming that all the results were consistent irrespective of the data sets, we only showed the results using (i) r, 70%, and single SNP per locus. For all of the above analyses , we set the following parameters: the minimum number of populations that a locus must be present in to process a locus (p)′ = 1, the minimum minor allele frequency required to process a nucleotide site at a locus (min_maf) = 0.01, the maximum observed heterozygosity required to process a nucleotide site at a locus (max_obs_het) = 0.9. BayeScan v 2.0 was used to detect possible SNPs under natural selection assuming two morphological species with a default setting.

A Bayesian individual-based assignment approach as implemented in STRUCTURE 2.3.4 was used to examine the genetic boundaries between P. elatius and P. konojoi individuals. Twenty independent runs were performed in STRUCTURE using an admixture model and allele frequency correlated model without any morphological priors. Both the length of the burn-in period and the number of Markov Chain Monte Carlo analyses (MCMC) were 200,000. We estimated lrtri K (Evanno, Regnaut & Goudet, 2005), the most likely number of clusters using STRUCTURE HARVESTER (Earl & vonHoldt, 2012), CLUMPAK (Kopelman et al., 2015), and DISTRUCT (Rosenberg, 2004) to summarize and visualize the STRUCTURE results. In addition, individual-based principle coordinate analysis (PCoA) was performed using GeneAlex ver. 6.5 to visualize the genetic relationship among different individuals in 2 dimensions. Data files were converted to each software using PGDspider ver 2.0.8.3 (Lischer & Excoffier, 2011).

Results

MtDNA analysis

A total of 961 bp (456 bp IGR1 sequences and 505 bp MutS after trimming unreliable sequences) from 25 individuals (13 P. elatius and 12 P. konojoi) were obtained and concatenated to reconstruct the phylogenetic tree (Fig. 1). The genetic diversity of the concatenated sequences for P. elatius and P. konojoi was low, with only three haplotypes with two polymorphic sites, including one gap, uncovered across 25 sequences in the concatenated sequence. The haplotype diversity (h) was 0.453 and nucleotide diversity (π) was 0.00047. The haplotype network analysis revealed P. elatius and P. konojoi shared a major haplotype (Fig. 1A). Four out of 12 P. konojoi individuals shared a major haplotype with P. elatius and all the P. elatius individuals shared the major haplotype with P. konojoi. The reconstructed phylogenetic tree indicated that the two morphological species are not monophyletic (Fig. 1B). P. konojoi included one polymorphic site and one gap site in IGR1 and no polymorphic sites in MutS regions, while All P. elatius shared a single haplotype at both loci.

Figure 1 MtDNA haplotype network considering gap regions (A) and maximum likelihood phylogenetic tree excluding gap regions (B).

P. elatius (PE) is shown in pink and P. konojoi (PK) is shown in black/white. (A) The size of the circle represents the number of haplotypes found in the analysis. (B) Only the nodes with bootstrap values (>50) were indicated.

MIG-seq analysis

In total, 25,444,986 raw reads with an average of 292,471 reads per sample were obtained for 87 individuals by MIG-seq analysis, of which 25,241,178 reads remained after filtering out low quality reads. We obtained 16,011,472 reads with an average of 184,039 reads per individual after two step filtering. The three SNP data sets created using different criteria resulted in a different number of loci: (i) r = 0.7 and only a single SNP per locus resulted in 223 loci, (ii) r = 0.5 and only a single SNP per locus resulted in 762 SNPs, and (iii) r was set at 0.5 and all SNPs per locus resulted in 2251 SNPs. All data sets indicated consistent STRUCTURE and PCoA results, regardless of the different ratio of missing data and possible linkage disequilibrium (Supplemental Information 1).

BayeScan indicated that all the loci were neutral (q-values >0.05). Among the 223 SNPs examined, 12 SNPs indicated species-specific alleles (fixed substitution between the species) that could be observed.

Both STRUCTURE and PCoA indicated the same patterns in all of the SNPs sets; P. elatius and P. konojoi are genetically distinct. Calculation of lrtri K of the STRUCTURE results indicated that K = 2 best explained the data with a mean likelihood = − 5, 666.970 and mean similarity score among 10 independent runs = 0.999. Clear genetic differences corresponded with the morphological differences without any genetically admixed individuals (Fig. 2A). PCoA indicated two morphological species are clearly separated by the x-axis, which explains 18.5% of the data (Fig. 2B).

Figure 2 STRUCTURE (A) and PCoA result (B).

(A) K = 2 with Mean (LnProb) = − 5666.970 and Mean (similarity score) among 10 runs = 0.999 and PCoA results using 223 SNPs data obtained by MIG-seq analysis (B). X axis indicates 18.5% and Y axis indicates 3.52% of the data. PK and PE represent P. konojoi and, P. elatius respectively.

Discussion

Japan has been the main exporter of the precious corals to mainland China. The increased demand of the precious corals in mainland China in recent years resulted in elevated prices for raw corals, and promoted illegal fishing and overexploitation. Precise harvest data for precious corals is currently unavailable due to the inconsistent application of CITES listings, unreported trade as personal or household effects, and illegal and unreported trade. Under these circumstances, having species-specific customs codes for CITES would be necessary to obtain accurate trade data for conservation (Shiraishi, 2018).

Previously, P. elatius and P. konojoi were considered morphologically different; traditional genetic analysis using mtDNA and nuclear elongation factor region (Tu, Dai & Jeng, 2015) did not indicate their monophyly and thus they have been genetically indistinguishable. In the present study, we confirmed that P. elatius and P. konojoi share a same haplotype (Fig. 1), which is consistent with the previous study. MIG-seq analysis, however, successfully discriminated P. elatius from P. konojoi. PCoA and STRUCTURE results indicated that the two species are clearly different with no intermediate genotypes, indicating no current hybridization between the two species. It is possible that extremely slow substitution rates of mtDNA of P. elatius and P. konojoi resulted in non-reciprocal monophyly due to the maintenance of an ancestral shared variation presented at the time of divergence. On the other hand, it is possible that a strong purifying selection could encourage them to split to reciprocal monophyly in nuclear DNA, as the MIG-seq analysis suggested more than 5% of the loci showed fixed substitution between P. elatius and P. konojoi, while BayeScan indicated all MIG-seq loci are all neutral.

This study demonstrated that P. elatius and P. konojoi are indeed different species without any hybridization or on-going gene flow consistent with morphological differences, and therefore require separate conservation management plans since P. elatius populations cannot be replenished by P. konojoi populations. P. elatius made up 20.6% of the total catch of precious corals in Japan from 1904 to 1920. Currently, this has dropped to 2.0% because the biomass of P. elatius has been dramatically depleted due to overfishing (Iwasaki, 2018); thus, there is an urgent need to devise and implement conservation strategies for P. elatius.

This study highlighted the effectiveness of the MIG-seq analysis using the high throughput sequencing technology for delimiting octocoral species. This study, as well as a previous study on shallow water octocoral species, demonstrated that MIG-seq analysis can uncover genetic lineages that are undetectable when using mtDNA or ITS2 (Richards et al., 2018). MIG-seq provides a time-saving (3 days at shortest), simple (two PCR steps), and economical (15 US dollars per sample) approach that is applicable even for small amounts of degraded or valuable samples for SNP genotyping. Data comparison among different species or genera is easy as the MIG-seq always uses the same multiplex primer sets, which would also be suitable for nuclear barcoding among different species. Further application of MIG-seq analysis to a wider range of octocoral taxa would help to clarify the unsolved status of closely related species.

Conclusions

This study indicated that P. elatius and P. konojoi are different species and that both species should be conserved separately. The results highlighted the importance of conservation of these two species, especially P. elatius, which has been dramatically depleted in the past 100 years. Again, it does suggest that separate conservation strategies will be required for the two different morphospecies. This study also demonstrated the effectiveness and robustness of MIG-seq for delimiting closely related octocoral species that were previously indistinguishable when using traditional genetic markers (mtDNA and EF). Further application of MIG-seq analysis for a wider range of octocoral or other taxa would aid in verifying the robustness of MIG-seq analysis on species delimitation.

Supplemental Information

Supplemental Information 1 MIGseqDatafile

Click here for additional data file.

Supplemental Information 2 MIGseqDatafile

Click here for additional data file.

Supplemental Information 3 FASTA file mtDNA

Click here for additional data file.

Supplemental Information 4 MIGseqDatafile

Click here for additional data file.

Supplemental Information 5 MIG-seq analysis using different SNPs set

Click here for additional data file.

Supplemental Information 6 Accession numbers

Click here for additional data file.

We are grateful to Mr. Masao Nakano, Yoshihiko Niiya, Katsuhiro Fujita for collecting samples in Kochi. We thank Mr. Hideaki Yuasa, Ms. Akemi Yoshida for their help in the analysis. Computations were partially performed on the NIG supercomputer at ROIS National Institute of Genetics.

Additional Information and Declarations

Competing Interests

Author Contributions

Field Study Permissions

Data Availability

The authors declare there are no competing interests.

Kenji Takata performed the experiments, analyzed the data, prepared figures and/or tables, authored or reviewed drafts of the paper, approved the final draft.

Hiroki Taninaka analyzed the data, approved the final draft.

Masanori Nonaka and Fumihito Iwase conceived and designed the experiments, contributed reagents/materials/analysis tools, approved the final draft.

Taisei Kikuchi, Yoshihisa Suyama and Satoshi Nagai analyzed the data, contributed reagents/materials/analysis tools, approved the final draft.

Nina Yasuda conceived and designed the experiments, contributed reagents/materials/analysis tools, authored or reviewed drafts of the paper, approved the final draft.

The following information was supplied relating to field study approvals (i.e., approving body and any reference numbers):

Samples were collected under the Kochi Prefecture sampling permit numbers Sa 401, Sa 412, and Sa 423. Other samples were from sample collections in Okinawa Churaumi Aquarium.

The following information was supplied regarding data availability:

All specimens and the accession numbers are available in the Supplemental files. mtDNA data is available at DDBJ: LC464485-LC464516–LC475110-LC475134. MIG-seq data is available at NCBI: PRJDB8153, BioSample: SAMD00166505–SAMD00166591, and DRA: DRX168319-DRX168404 (Experiment), DRR177792-DRR177878 (RUN): https://ddbj.nig.ac.jp/DRASearch/submission?acc=DRA008354.

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
