# Peer review of "Multiplexed ISSR genotyping by sequencing distinguishes two precious coral species (Anthozoa: Octocorallia: Coralliidae) that share a mitochondrial haplotype"

_PeerJ, doi:10.7717/peerj.7769_

## Round 0.1 · original submission · Minor Revisions

Reviewers have raised that some aspects of the papers require attention. I encourage the authors to address all the comments and suggested modifications provided by the reviewers.

·

Basic reporting

Although the text is for the most part clearly understandable, there are nonetheless a few places where it might benefit from the attention of an English-language editor. There are some minor grammatical errors, and in a few places missing words or non-standard grammatical constructs make a sentence difficult to understand. I have noted the most serious of these in the general comments section. There are also places where additional citations would be helpful. In particular, citations are missing for some of the software or scripts used for data analysis.

More seriously, the figures (as reproduced in the reviewer's copy) are far too small to read, and when enlarged their resolution is very poor - as a result I cannot read any of the labels, and individuals cannot be discerned in the STRUCTURE plot. In addition, the acronyms used for species seem to differ between the two figures and between the legends and figures (PK, PE vs. CK, CE). It may be that the figures are adequate for publication (if acronyms are fixed), but I can't really evaluate them from the versions that have been provided to me. Finally, it does not appear that the raw data files (sequence reads for the MIG-seq) have been provided. There is a supplemental data file that includes "MIGseqDatafile"in its name, but it is a large spreadsheet of integers with no heading to indicate what the data represent (alleles? SNPs?). Ideally, in order to be able to reproduce the results of the analyses, readers should have access to the original sequence read files (deposited to a public repository such as GenBank's Sequence Read Archive (SRA) or international equivalent).

Experimental design

This appears to be a well-designed study with results that are unambiguous and easily interpretable. This is (to my knowledge) only the second use of MIG-seq to delimit species in corals (by the same research group), and it seems to be a very useful tool to apply in cases where conventional barcode markers have failed to discriminate morphospecies. Because this technique has not been used widely (compared to RADseq), it would be helpful if a bit more background were included for the reader who is unfamiliar with the definition of an ISSR and the basic concept of the MIGseq approach. Citations of some additional references (for instance the original Suyama paper) at first mention of the technique would be appropriate. The Richards et al. paper included a very clear, concise description of MIGseq (in both the Intro and Methods sections), and I would suggest including similar language in this manuscript. The steps in data analysis seem to be described clearly, but as already mentioned above, citations are missing for some of the programs used.

Validity of the findings

The results are presented clearly and appear to show unambiguous genetic differentiation of the two Pleurocorallium morphospecies. There were, however, a number of different analyses that were run using different parameters (for example, different amounts of missing data and SNPs per locus), but the results of only one parameter set are shown. Instead of just stating "data not shown", it would be very appropriate to include the results of the analyses with other parameter sets as supplemental figures that could be viewed and assessed independently by the interested reader. The conclusions are sound, but I suggest wording them a little differently: this study doesn't highlight the importance of conservation of these species per se, but it does suggest that separate conservation strategies will be required for the two different morphospecies.

Additional comments

53: I think it would be clearer in this sentence to specify "family Corallidae" rather than "coraliid corals".

81-83: Your statement that no one has used high-throughput sequencing methods in precious corals is incorrect. Herrera & Shank (2016, MPE 100: 70-79) used RADseq to delimit species in Paragorgia- the sister family to Corallidae - and they also included Corallium and Hemicorallium in their study as outgroup taxa. It would be appropriate to acknowledge and cite their study.

96: This sentence is confusing. Rather than "have at least two distinct morphological features" I think what you mean is "can be distinguished by at least two morphological features..."

100-101: This sentence is not clear and it seems like there might be one or more words missing. Should it say that the two species *shared haplotypes* at eight mtDNA regions and EF-1?

104: The meaning of the phrase "whether they are the same species with the current gene flow" is unclear. Perhaps this should read "whether they are the same species with gene flow occurring between them"?

120-121: The formal name of mtMutS is the "mitochondrial mutS-like protein". Note, however, that it has still not definitely been shown to be operating as a mutation-repairing gene in the octocoral mt genome. That function is still speculative.

190: The meaning of the statement "P. konojoi had only one haplotype in P. konojoi" is not clear.

Table 1: Should "submatrine" be "submarine"?

201: The correct spelling is disequilibrium, not disequibilidium.

·

Basic reporting

The mtDNA of anthozoans accumulates nucleotide substitutions at far slower rates than do bilaterally symmetrical animals. These slow rates have made the inference of phylogeographic structure and distinguishing closely related taxa far more difficult for anthozoan taxa than most other animals. The authors here employ a newly developed method to generate nuclear SNPs to tell apart two closely related octocorals that are commercially traded and may face threats to their populations.

The manuscript has two figures and both could be improved. Figure 1 needs a major reworking because it does not convey the critical information that it should (that mtDNA cannot distinguish the two morphospecies). Figure 1 should be a haplotype network instead of a rooted phylogenetic tree. The long branch to the outgroup makes it impossible to see the limited mitochondrial variation that the figure should focus on. In addition, the Fig. 1 legend should state the type of data (mtDNA) that the tree/network is based upon and the gene region as well. Figure 2 is generally good, but could be improved by spelling out the two species' names in Fig 2a so the reader does not have to recall the abbreviations.
The background material presented in the Introduction is too narrow. The slow mtDNA issue has been long-recognized for anthozoans (see Hellberg 2006 BMC Evol Biol 6:24, see also octocoral work by SC France) but focus here is not just octocorals, but even more specifically on coralliids. The broader context of slow mtDNA in anthozoans should be provided by including some more of these references.
The English used is generally intelligible (other than in parts of the Discussion, see below), but awkward. I note some suggested changes here, but the manuscript deserves a more thorough editing by a native speaker familiar with the subject.
Line 37: Indicate how many of each species here rather than the total for both combined.
37-38: "from the Ryukyu Islands and Shikoku Island..." ["to" implies that there are multiple sampling locations between the two named extremes, when there are actually just the two collecting sites].
53: "Precious, or coralliid, corals (Anthozoa: Octocorallia) play an ..."
64: Reference needed here
87-93: Any evidence that this approach identifies single-copy loci?
99-102: A critical sentence that need clarifying. What PROPORTION of haplotypes are shared between the two? Are they the most common ones?
141: Says 87 in the abstract and elsewhere. Why the difference?
183: Need to indicate how many of each species here, not just the total for the two combined.
189: There is no Fig. 1b
205: "either" unclear when there are three alternative data sets, not two
240: "composed" instead of "occupied approximately"

Experimental design

Meets standards - no comment

Validity of the findings

Their central conclusion, that the MIG-seq approach they have taken allows two potentially threatened precious corals to be distinguished even though mtDNA cannot do the same, is valid. Some ancillary points made in the Discussion, however, confused me. Generally, the writing in the Discussion was the hard to fathom. Minimally, there appears to be confusion regarding what lineage sorting is and how it relates to the results they have obtained. Lineage sorting depends on the effective population size, which is smaller for mtDNA than for nuclear DNA. Thus, lineage sorting will be faster for mtDNA, contra what is stated in line 225. If the two species diverged while sharing a single common mtDNA haplotype, then slow substitution rates would mean that it takes a long time for differences to accumulate between the genetically isolated lineages. This is NOT lineage sorting, however, which refers to non-reciprocal monophyly due to the maintenance of shared variation present at the point of divergence. Selection at nuclear loci, including the 12 diagnostic SNPs identified here, could speed them to reciprocal monophyly.

Additional comments

Nothing additional

---

## Round 0.2 · Minor Revisions

One of the reviewers have provided a few further, constructive suggestions and I encourage the authors to address them and resubmit.

·

Basic reporting

The authors have improved the English and addressed a majority of the reviewers' comments to clarify what were previously confusing statements. The figures have also been improved significantly in size, resolution and readability. There are, however, still a number of places throughout the manuscript where minor grammatical errors or word choice make the meaning of a sentence unclear. These include:

70: The scientific definition of "precious corals" is still not clear. The best way to state this would be as in the paper's title, i.e. "The precious corals (Anthozoa: Octocorallia: Coralliidae), "

157: This should read "using other techniques such as RAD-seq"

159: " the method can be performed on few or low-quality DNA" is unclear because "few" is not an appropriate modifier for "DNA". Maybe this should read "the method can be performed with small amounts or low-quality DNA"? or "with low concentration or low-quality DNA"?

162: The statement "Octocorallia, which is closely related to Heliopora spp," makes no sense (Octocorallia is the group to which Heliopora belongs). In the context of the previous study, what would make sense to say here is "Closely related species of Heliopora were successfully delineated"

307: The statement "P. konojoi shared one haplotype in P. konojoi" has been changed in response to previous comments, but still makes no sense. Should this say something like "All P. konojoi shared a single haplotype at both loci"?

332-333: To be consistent with the Introduction, Corallium (a specific genus) should be replaced with either "coralliid" or "precious" in both of these sentences.

396: "among different species or genus" should be "among different species or genera"

The Figure legends need to explain in more general terms what is shown, rather than just have telegraphic titles such as "Structure". The figures should stand on their own and be understandable without the reader having to refer to the text to know, for example, what species or gene(s) have been analyzed.

The text states that samples were collected from the Ryukyus and Shikoku islands, but Table 1 does not indicate what collection sites are in the Ryukyus and which are at Shikoku.

Fig 1, a: What is the red circle and the number 43 along the line connecting it to the pink circle? Is that the outgroup? If there were only two haplotypes found among P. konojoi and P. elatius a haplotype network is not very informative. I recommend leaving it out and showing only the ML tree.

Experimental design

Reviewers' suggestions have been addressed, and appropriate references have been added.

Validity of the findings

Reviewers' comments have been addressed, and a supplementary file has been added to show the results of the Structure analyses run with different parameters.

Additional comments

Overall, the authors have done a thorough job of addressing the reviewers' comments and suggestions. Just a few more minor corrections are necessary to make the text as clear as possible.

·

Basic reporting

The English has improved. Most notably, the figures are much improved.

Experimental design

Sound, as before

Validity of the findings

The previously confusing mentions of lineage sorting have been clarified.

Additional comments

I am happy with the revisions and recommend acceptance.

---

## Round 0.3 · accepted · Accept

The authors addressed satisfactorily all comments and suggestion provided by the reviewers. So the current version is ready for publication.